Rituals decrease the neural response to performance failure

Hobson Nicholas M. nick.hobson@utoronto.ca 1
Bonk Devin 1
Inzlicht Michael 1 2
1 Department of Psychology, University of Toronto , Toronto , Ontario , Canada
2 Management, Rotman School of Management , Toronto , Ontario , Canada
Hung Tsung-Min
Electronic publication date: 2017 May 30
Publication date: 2017
Volume: 5
Electronic Location ID: e3363
Received 2017 Mar 7; Accepted 2017 Apr 27
Copyright: ©2017 Hobson et al.
Copyright year: 2017
Copyright holder: Hobson et al.
License: This is an open access article distributed under the terms of the Creative Commons Attribution License, which permits unrestricted use, distribution, reproduction and adaptation in any medium and for any purpose provided that it is properly attributed. For attribution, the original author(s), title, publication source (PeerJ) and either DOI or URL of the article must be cited.
License URL: https://creativecommons.org/licenses/by/4.0/

Keywords: Ritual, Self-regulation, Error-related negativity, Neural performance-monitoring

Funding: Social Science and Humanities Research Council This work was supported by a Social Science and Humanities Research Council grant to NM Hobson and M Inzlicht. There was no additional external funding received for this study. The funders had no role in study design, data collection and analysis, decision to publish, or preparation of the manuscript.

==============================
Rituals are found in all types of performance domains, from high-stakes athletics and military to the daily morning preparations of the working family. Yet despite their ubiquity and widespread importance for humans, we know very little of ritual’s causal basis and how (if at all) they facilitate goal-directed performance. Here, in a fully pre-registered pre/post experimental design, we examine a candidate proximal mechanism, the error-related negativity (ERN), in testing the prediction that ritual modulates neural performance-monitoring. Participants completed an arbitrary ritual—novel actions repeated at home over one week—followed by an executive function task in the lab during electroencephalographic (EEG) recording. Results revealed that relative to pre rounds, participants showed a reduced ERN in the post rounds, after completing the ritual in the lab. Despite a muted ERN, there was no evidence that the reduction in neural monitoring led to performance deficit (nor a performance improvement). Generally, the findings are consistent with the longstanding view that ritual buffers against uncertainty and anxiety. Our results indicate that ritual guides goal-directed performance by regulating the brain’s response to personal failure.

Rituals Decrease the Neural Response to Performance Failure

Rituals are human universals. As broad behaviors existing across multiple domains, rituals can be thought of as formal sequences characterized by rigidity and repetition that are embedded in a larger system of meaning. In ritual, ordinary sequenced actions become transformed, like when an athlete dons equipment from bottom-to-top (but not top-to-bottom) as a pregame ritual. A puzzling feature of many rituals is that they require a person to invest time and energy into completing the actions, often with immediate instrumental value. In a way then, rituals pose an economic cost problem (Irons, 1996): why do people engage in these behaviors—often repeatedly, and over a lifetime—if they reveal no direct benefit to the self?

Performance rituals may offer insight into this question. Research and anecdotal evidence suggests that these types of rituals influence goal-regulated behaviors by heightening motivation and minimizing extraneous sources of anxiety. Research to date has mostly investigated pre-established rituals, in which the actions are tied to broader personal meaning, precluding us from knowing the precise causal impact of the ritual itself. Thus, to examine ritual’s underlying causal properties, we use novel, lab-created ritual actions and control for the pre-established features tied to the broader ritual context. In a fully pre-registered experiment (https://osf.io/hcmkp/) we directly examine the impact of arbitrary rituals on the task performance and on the associated brain-based regulatory system.

Ritual and goal-directed performance

Rituals often precede important tasks and performance events, such as games, sports competitions, military operations, presentations, and tests. Ethnographic research has documented the importance of these rituals: anthropological evidence from ancestral cultures finds that taboo rituals—those which prescribe rules of conduct and order—tend to increase during times of dangerous hunting/fishing performances (Malinowski, 1954; Sosis, 2000); professional and amateur athletes feel compelled to do the exact same warm-up ritual before every game (e.g., Rotella & Cullen, 1995); military culture espouses disciplined excellence through ritualistic training programs (e.g., King, 2006; McNeill, 1997); and even popular blog posts, with titles like “The Value of Ritual in Your Workday” (Bregmen, 2010), give readers advice on how to utilize rituals for achieving small, day-to-day goals (e.g., Reynolds, 2011). Despite the ubiquity of performance rituals, very little is actually known about how these behaviors influence goal-directed performance, if at all.

In performance domains, most notably in sports athletics, there is some evidence showing rituals serve a regulatory function. For instance, performing rituals during athletic events, like right before shooting a free-throw in basketball, helps players perform better (Czech, Ploszay & Burke, 2004; Lobmeyer & Wasserman, 1986; Predebon & Docker, 1992), especially in high-stakes competition and stress (Gayton et al., 1989; Wrisberg & Pein, 1992). The thinking is that rituals help improve performance because they mobilize motivational and regulatory states, either through improving concentration, creating physical readiness, or boosting confidence (Foster, Weigand & Baines, 2006; Weinberg, Gould & Jackson, 1979).

This raises an important question of how pre-performance rituals differ from similar preparatory routines/habits. A wealth of research on pre-performance routines in athletes suggests that these behaviors also act as a regulatory mechanism for controlling arousal, attentional focus, and task expectations (e.g., Cohn, Rotella & Lloyd, 1990). Despite serving similar regulatory functions, rituals are in fact considered distinct from habits or routines. Rituals possess a high level of rigidity and formality, with steps occurring in a fixed episodic sequence (Foster, Weigand & Baines, 2006; Rook, 1985; Rossano, 2012). In addition, unlike habits or routines, which have measurable utility in achieving a goal-state and are context-dependent (e.g., Wood, Quinn & Kashy, 2002), rituals appear to have no direct function. The causal connection between ritual and the attendant goal is less transparent than other instrumental behaviors like habits, a distinction often referred to as the ritual stance versus instrumental stance (Herrmann et al., 2013; Kapitány & Nielsen, 2015; Kapitány & Nielsen, 2016; Legare et al., 2015).

Consider the example of an Olympic athlete whose pre-game ritual is to place her shin pads in a repeated manner from bottom-to-top. This fixed set of actions, if done properly, makes her feel that she is ready to perform her best. But if she alters the sequence, placing pads instead from top-to-bottom, she may feel the preparation is “off” and, thus, her performance will suffer. Although the actions in this case are arbitrary, and the order of pad placement has no direct impact on performance (i.e., the behavior lacking overt instrumental value), the sequenced repetition of the ritual might leave the athlete feeling calm and prepared. In contrast, a pre-performance routine might involve similar actions, but the element of scrupulously adhering to a particular sequence falls away; sticking to the script is no longer imperative as it is during ritual (Dulaney & Fiske, 1994). Here the athlete’s routine might entail putting on her pads, but without feeling compelled to do the preparatory actions in any particular prescribed order. Thus, pre-performance rituals possess a deep-level irrationality, but even still, they appear to promote effective goal-pursuit. How does this happen? Recent experimental evidence from social and cognitive psychology has begun to shed light on the underlying processes involved.

For instance, it has been shown that activating superstitious concepts—irrational cognitions frequently found in different rituals—leads to improved performance by making people feel like luck is on their side, thereby boosting feelings of self-efficacy and confidence (Damisch, Stoberock & Mussweiler, 2010; but see Calin-Jageman & Caldwell, 2014 for a failed direct replication of these effects). Like superstitious thinking, rituals may improve performance by fostering personal confidence and positive performance expectations, which in turn, leads to greater success on the ensuing task. This process may act like a placebo: If a person believes a ritual will help their performance, then having that belief alone will increase the chances of success.1

A very recent set of studies demonstrated that people tend to perform better on a range of tasks when engaging in an arbitrary ritual beforehand, but only when the actions are labeled as ritual. The researchers found evidence that participants’ improved performance was explained by a reduction in self-reported anxiety (Brooks et al., 2016). It has long been inferred that rituals—particularly religious rituals—can act as a palliative against anxiety and failure. Indeed, evidence suggests that the completion of chunked and predictable action sequences of a personal ritual can act as a compensatory strategy to help regain control when experiencing anxiety (Boyer & Liénard, 2006; Hirsh, Mar & Peterson, 2012; Norton & Gino, 2014). But certain rituals may be better equipped than others for this. For example, Reinvestment Theory (Masters, 1992; Masters & Maxwell, 2008) suggests that making relatively automated movements more consciously active could disrupt the associated motor program. Given this, an effective ritual might be one that prevents against reinvestment by relinquishing conscious control by automating predictable sequences (Jackson & Masters, 2006).

But how do such arbitrary sequenced behaviors impact goal-regulation, performance, and anxiety? What neuropsychological mechanisms underlie these types of rituals? To gain insight into this puzzling regulatory process, we can consider how the underlying mechanisms of ritual are driven by brain-based regulatory control, specifically the neural performance-monitoring system.

Neural performance monitoring, the ERN, and ritual

As a candidate proximal mechanism underlying the ritual-performance link, we expect to see changes in neural performance-monitoring, especially when considering its involvement in tracking motivationally relevant performance outcomes. Here, we wondered whether ritual’s regulatory function can be traced either to heightened performance-monitoring (an orienting to failure), or to reduced performance-monitoring (minimizing reactivity to failure).

Offering a test of these predictions, when a person experiences some sort of performance failure, they tend to exhibit a distinct neurophysiological response, which indicates that such failures are not only monitored but also responded to emotionally (Inzlicht & Al-Khindi, 2012) and perhaps with personal distress (Hajcak & Foti, 2008). A common neural correlate of this monitoring system is the error-related negativity (ERN), an electroencephalographic waveform believed to arise from the dorsal region of the anterior cingulate cortex (Bush, Luu & Posner, 2000; Dehaene, Posner & Tucker, 1994; Gehring et al., 1993; Mathalon, Whitfield & Ford, 2003). As a generic, multi-modal signal of the performance-monitoring system, the ERN is thought to be related to reinforcement learning (Holroyd & Coles, 2002) and conflict monitoring (Botvinick et al., 2001; Yeung, Botvinick & Cohen, 2004), reflecting the discrepancy between an expected outcome and an actual outcome (Holroyd & Coles, 2002). It emerges between 50 and 100 ms after an error is committed.

Importantly, recent work has shown that the ERN may reflect not only the detection of errors, but also the affective and motivational salience of performance failure, error, and conflict (e.g.,  Hajcak et al., 2005; Hobson et al., 2014; Inzlicht, Bartholow & Hirsh, 2015; Proudfit, Inzlicht & Mennin, 2013; Weinberg, Riesel & Hajcak, 2012). According to this view, the signal strength of the ERN may partially reflect an evaluative appraisal of error or response conflict, with differences in ERN magnitude mapping onto the sensitivity to performance failure and regulatory control.

Facilitating effective control during performance, the ERN is thought to be involved in the detection of conflict/error (e.g., Botvinick et al., 2001; Falkenstein et al., 2001), the openness to experiencing aversive performance error (e.g., Legault & Inzlicht, 2013; Teper & Inzlicht, 2013), and the affective or evaluative response to that performance error (e.g., Bartholow et al., 2012; Hajcak & Foti, 2008; Saunders, Rodrigo & Inzlicht, 2016). Critical to the test of our current hypotheses, we can map the differences in ERN activation onto the explanation of ritual’s predicted regulatory function. On the one hand, larger ERNs tend to reflect heightened performance and error monitoring (Van Veen & Carter, 2002), an internally driven mechanism optimizing performance by signaling the need for remedial control (Carter et al., 1998). If rituals evoke a sense of confidence and self-efficacy, then we should see an increase in performance-monitoring with increased motivation to detect potential sources of failure (i.e., a larger ERN).

On the other hand, smaller ERNs tend to reflect a blunted affective response to personally distressing information like errors and performance failures (Maier et al., 2016), the opposite being expressed in people with anxiety disorders and high trait anxiety/worry (Hajcak, McDonald & Simons, 2003; Olvet & Hajcak, 2009a; Olvet & Hajcak, 2009b; Weinberg & Hajcak, 2011). If, as some evidence has suggested, rituals appear capable of reducing anxiety and mitigating uncertainty, then it is equally plausible that ritual leads to a decrease in performance-monitoring (i.e., a smaller ERN). A possible downside, of course, is that this type of blunted neural evaluation of errors might end up hurting performance rather than helping it (e.g., Hobson et al., 2014; Ridderinkhof et al., 2002).

The current study

Here, in our openly pre-registered experiment (https://osf.io/hcmkp/), we asked if even arbitrary ritual can influence the ERN, and if so in which direction (amplified or muted ERN). Do rituals regulate goal-directed performance by (i) improving regulatory control through enhanced error/failure detection, or (ii) dulling the threat of personal failure? For support of (i), we would expect that rituals would lead to heightened performance-monitoring, as shown by an amplified ERN. For support of (ii) we would expect that rituals would lead to a reduction in performance-monitoring, as shown by a muted ERN.

Methods

Participants

A power analysis using G*Power (Faul et al., 2007) was run to determine sample size. Using a mixed, pre-post design and assuming the typical moderate effect size in social psychology (i.e.,  r = .21, d = .43; Richard et al., 2001) and high correlations between repeated measures typical of event-related potential (ERP) designs (r = .60–.80; Olvet & Hajcak, 2009a; Segalowitz et al., 2010), we determined that a total sample size of 48 participants yields a power value of 0.90 for detecting the hypothesized within-between interaction. We therefore decided to stop data collection between 50–60 participants to account for participant exclusions. In line with open science practices, we pre-registered the experiment on Open Science Framework (https://osf.io/hcmkp/), where we outline the question of the two possible explanations, the associated predictions, and detailed procedures.

Fifty-nine participants were recruited from the University of Toronto Scarborough Campus through introductory psychology courses and campus advertisements. All participants were provided with a consent form and signed their written consent to participate. Eleven participants were excluded from all analyses due to equipment/hardware malfunction (n = 3), high EEG artifact rate (>30%; n = 5), failure to understand task instructions (n = 1), and the commission of too few errors to calculate reliable evoked potentials (n = 2). ERN calculations were based on no fewer than five artifact-free error trials (Olvet & Hajcak, 2009a; Olvet & Hajcak, 2009b). Half of one participant’s data was excluded because they made too few errors in the second block of the in-lab experiment (fewer than five). Their usable data was still entered into the mixed effects models using MLM (more below; see Gueorguieva & Krystal, 2004). This left us with a final sample of 48 participants (35 females; mean age = 19, SD = 1.48). Across conditions (ritual condition n = 22, control condition n = 26), participants were told that they could earn bonus money based on how well they completed the task ($10 guaranteed, with a chance to earn a bonus $10). However, regardless of how they performed, all participants received $15 for participating (as well as a course credit for those in introductory psychology courses). This deception (in regards to monetary compensation) was used to ensure that participants were motivated to perform at their best during the performance tasks. The University of Toronto granted ethical approval to carry out the study on human participants within its facilities (protocol reference #28411). Participants gave full written consent before the start of the experiment.

Procedures

The experiment was comprised of an at-home and in-lab portion, lasting seven days in total. Participants were randomly assigned to a ritual or control (non-ritual movement) condition. They were told that the purpose of the study was to examine the mind-body connection and the effects of physical movement on cognitive performance. Importantly, the purpose of the study was kept vague in order to hold it constant across the two conditions and to control for study demand characteristics.

In the ritual condition, participants were asked to learn and memorize a set of elaborate ad-hoc action sequences (see Table 1 for specific steps of the ritualized actions). The operationalization of ritual consisted of these novel and arbitrary actions, which were meant to mirror the features of real-life rituals. These features included repeated and sequenced movements accompanied by a set of instructions that were prescriptive and rule-bound (for similar manipulations, see Norton & Gino, 2013; Hobson et al., in press). Importantly, the term ritual was never mentioned to participants, instead merely referring to the behaviors as action sequences; this was done to control for any prior beliefs participants may have had about rituals and the expectations of what rituals do to assist performance. Participants were provided with written instructions of the sequenced steps as well as a video model (https://osf.io/u2t75/) to help them follow along and commit the actions to memory.

Table 1 The specific actions that were given to participants during the at-home portion of the study.

The behaviors were precisely ritual-like with features mirrored real life rituals, including an emphasis on fixed episodic sequences and repetitive movements.

Action sequences instructions	
1. Place your hands flat on the table. Spread your fingers of your left hand and keep your fingers of your right hand together. Close your eyes, and take three slow deep breaths.	
2. Now switch: spread your fingers of your right hand and bring your fingers of your left hand together. Close your eyes, and take three slow deep breaths.	
3. Place the backs of your hands on the table, close your eyes, take three slow deep breaths.	
4. Make fists and turn your fists over so the top of your wrist is facing upwards and your thumbs are inwards towards each other. Close your eyes, take three deep breaths.	
5. Bring your fists together at your chest, slowly raise them above your head, and as you do draw in a large inhale through your nose. Return your fists to your chest while drawing out an exhale through your mouth. Repeat this three times.	
6. Bring your open hands together at your chest (palm to palm), slowly raise them above your head, and then bring them back to your chest. Repeat this three times.	
7. With your palms together at your chest, interlock your fingers so that your dominant hand’s thumb is over the non-dominant thumb (i.e., dominant thumb touching your chest) and raise your hands above your head. Return them to your chest. Repeat this three times.	
8. Starting at your chest, bring your interlocked fists to your right shoulder and hold for three seconds. Bring your interlocked fists back to your chest and hold for another three seconds. Bring your interlocked fists to your left shoulder and hold for another three seconds. Return your fists to your chest and clap twice.	
9. Rest your hands, palms facing towards each other on the table in front of you. Hold for three seconds. Close your eyes and pull your hands apart. With your index/pointer fingers, tap the table ten times.	
10. You are now finished.	

In the control (non-ritual movement) condition, participants instead completed a coordinated movement task in which they were instructed to move their arms/hands at different times to touch moving block images on the computer screen. The amount of time and movement involved was the same as with the ritual actions. The only difference was that the control movements were not sequenced or repeated (i.e., strictly non-ritualistic). We were careful in this control manipulation so that we could infer that any observed differences would be attributable to the particular ritual movements themselves, and not any other non-ritual features of the at-home experience.

All participants were sent five email reminders, one per day, during the at-home portion and prompted to complete the questionnaire and tasks. On day 1 participants were recruited and introduced to the study and asked to complete a simple demographic questionnaire; on days 2–5 participants completed the main part of the at-home portion; on day 6, participants received feedback on their task performance over the past four days (more on this below); finally, on day 7 participants were scheduled for the in-lab portion of the study.

For the first four days, participants in the ritual condition were asked to complete the action sequences (following along with the model video for the first two days and then from memory for the remaining time), while participants in the control condition did the same randomized block touching task each day. Afterwards, all participants completed a short version of a Stroop task programmed using Inquisit 4.06. The Stroop task (Stroop, 1935) is a canonical inhibitory control task that reliably tracks executive function ability. It was used during the at-home portion in this case because we wanted participants to complete a reaction-time task that was similar to the task they would be completing in the lab, but still different in order to control for practice effects. Using a different (though related) executive function task allowed us to examine the ritual-performance link over the duration of the week-long experiment.

The Stroop task consisted of a series of colored words (red, green, blue, or yellow; congruent and incongruent trials) or colored blocks (control trials), in which participants were asked to identify the color in which each word was presented. Each trial consisted of a fixation cross (‘+’) presented for 500 ms, followed by the color word (block) presented for 200 ms, with a response window of 1,000 ms. Participants completed 84 trials, split equally with 28 congruent trials, 28 incongruent trials, and 28 control trials. Last, participants responded to eight items on a scale from 1 (strongly disagree) to 7 (strongly agree) asking how they felt about their performance on the Stroop. The questions read, “I felt [item here] about my performance”. The items included: accomplished, confident, able to handle, mastery, anxious, calm, in control, and frustrated. For the purposes of the current investigation, we do not evaluate or discuss the results of the at-home task and questionnaires.

On the final day of the at-home portion, all participants were sent an email containing false performance feedback from the previous four days. The feedback consisted of a line graph showing participants reaction-times and error rates broken down by day, in comparison to the running overall average of the study. The plotted lines showed that participants had improved over the week and had been performing above average. In reality, every participant in both the ritual condition and control condition received identical feedback. This was done in order to convince participants that their performance was improving over time, further incentivizing their performance on the upcoming laboratory task.

To ensure high compliance among participants during the at-home portion of the experiment, we reminded them that the full participation required completion of both the at-home tasks as well as the in-lab component. Our aim was to get participants to complete the ritual or control actions every day during the week, or as often as they possibly could. To help with this, we also tracked the number of times participants logged into the at-home survey, which included following the action instructions (and video for Days 1–3) and completing the Stroop task. Additionally, the final question of each survey asked participants to confirm that they completed the actions (ritual or control) as instructed. Because our experimental manipulation relied on this longitudinal experience of the at-home portion of the study, to include participants who failed to do the tasks only once (or not at all) would compromise the manipulation of our independent variable. Four participants in total (n = 2 in the ritual condition, n = 2 in the control condition) showed low compliance and completed the at-home task only once. They were excluded from completing the in-lab portion but still offered credit. Low compliance was also confirmed during a funnel debriefing at the end of the in-lab session, when participants reported the number of times they completed both the survey and actions (either ritual or control) over the course of the week. This revealed that most people did in fact follow the instructions and completed the actions almost every day of the at-home component (M = 6.78).

After one week, participants came into the lab to complete the final portion of the study. When they arrived, participants were seated in front of a computer station and prepared for EEG recording. Continuous recording EEG was used while participants performed two rounds of a speeded inhibitory control task in a dimly lit room set at a viewing distance of approximately 100 cm to the screen. Different inhibitory control and speeded reaction tasks have been shown to elicit a reliable ERN signal. The performance paradigm used here was a modified go/no-go task, which consisted of a two-alternative, forced-choice task, where participants were instructed to press the/key with the right-hand if they saw an M (“go”) target, and to press the Z key with their left-hand if they saw a W (“nogo”) target. Key assignment was set on a standard QWERTY format and responses recorded using a millisecond sensitive DirectIN keyboard (Empirisoft, New York, NY, USA). The M letter served as the frequent stimuli (.80 probability of appearing) and the W letter served as the infrequent stimuli (.20 probability of appearing). This imbalance in stimuli presentation was to ensure that participants developed a prepotent response tendency towards the frequent M (“go”) stimuli, thus making the infrequent W (“nogo”) stimuli error-prone, with successful button-press requiring inhibitory control. Because the task was forced-choice, both infrequent and frequent responses were recorded and analyzed. In the results section, we refer to the frequent and infrequent stimuli as low-conflict and high-conflict trials, respectively.

Each trial commenced with a centered fixation cross (randomly presented between 150–300 ms), followed by the presentation of the target letter (200 ms) in white font on a black background and centered in the middle of the screen. The screen then remained blank until one of the response keys (/or Z) was pressed or until the maximum response window (500 ms) was reached, in which participants were instructed to respond faster. The intertrial interval varied randomly between 450 and 1,000 ms before the start of the next trial. An incorrect response was immediately followed by an “error” message in white font on the black background and lasting for 500 ms before proceeding to the next trial.

To incentivize their performance, participants were given the following instructions: “The faster and more accurate you are in your responding, the more bonus pay you will be given. Remember, peak cognitive performance = fast & accurate”. Participants first completed 20 practice trials where they were told their performance was not being monitored for the bonus pay. Following this, they completed a total of 760 experimental trials, which were divided into 8 blocks of 95 trials and separated by self-paced rest periods in between. We used a pre/post within subjects experimental design, in which all of the participants first completed the first 4 blocks (i.e., pre-rounds), followed by the movement manipulation done once more in the lab, and then again the final 4 blocks (i.e., post-rounds; see Fig. 1 displaying the in-lab experimental design).

Figure 1 A pre/post mixed experimental design for the in-lab portion of the study.

Participants completed 380 pre-round trials, underwent the action manipulation (ritual versus control), and then completed 380 post-round trials. Motivated preparation and subjective experience (emotion and efficacy) were asked before and after each round.

Before beginning the pre-rounds, participants answered four motivated preparation questions, which asked them to think about and anticipate their upcoming performance (e.g., “How mentally prepared do you feel heading into the task?” and“ How well do you expect to perform on this round?”) on a 1 (“not prepared at all”; “not well at all”) to 7 (“very prepared”; “very well”) scale. The four motivated preparation items were combined into an aggregate score and used in subsequent analyses (Cronbach’s α = .87). After answering these questions, participants completed 4 blocks of the pre-round trials.

After the pre-rounds, participants were asked to report their subjective experience of how they felt during their performance using the same eight items from the at-home portion. Three questions asked about participants’ specific emotional experience (anxious, calm, and frustrated), while the remaining five questions asked about their self-efficacy (control, accomplished, confident, handle, and mastery). Anxiety and frustration were reverse-scored and combined with calm to create an aggregate positive affect score (Cronbach’s α = .86). Similarly, control, accomplished, confident, handle, and mastery were combined to create an aggregate score for self-reported efficacy (Cronbach’s α = .94).

The actions manipulation was carried out in between the pre- and post-rounds. Participants in the ritual condition were asked to complete the action sequences as they had memorized them over the course of the week, and to press the spacebar key once finished. Participants in the control condition completed the block touching task for the same amount of time. Afterwards, all participants completed the four motivated preparation questions like before. Participants then finished the final 4 blocks of the post-round trials, and again answered the same emotion and efficacy questions.

At the end of the performance task, participants in the ritual condition completed a set of questions that asked how they felt about their week-long experience in completing the action sequences. There were six items related to the ritual appraisal (ease, enjoyment, effort, special, meaningful, and powerful). We reverse scored effort and combined all the items (Cronbach’s α = .78) to create an index for positive ritual appraisal. Finally, participants in the ritual condition were asked to report how they felt about the experience of learning the action sequences over the course of the week. We recorded their open-ended answers for qualitative purposes. All participants were then thanked and fully debriefed.

Neurophysiological recording and analysis

Continuous EEG was recorded during the eight blocks of the executive function task using a stretch Lycra cap embedded with 32 tin electrodes (Electro-Cap International, Eaton, OH, USA). The scalp-electrode montage consisted of midline electrode sites (FPz, Fz, FCz, Cz, CPz, Pz, and Oz). The continuous EEG data were digitized using a sample rate of 512 Hz, and electrode impedances were maintained below 10 kΩ during recording. EEG activity were amplified using an ANT Refa8 TMSi device (Advanced Neuro Technology, Enschede, The Netherlands). Offline, both sets of signals were referenced analyzed with Brain Vision Analyzer 2.0 (Brain Products GmbH, Munich, Germany).

The EEG data was digitally filtered offline between 0.1 and 15 Hz and the data corrected for vertical electro-oculogram artifacts (Gratton, Coles & Donchin, 1983). An automatic procedure was employed to detect and reject artifacts. The criteria applied were a voltage step of more than 25 µV between sample points, a voltage difference of 150 µV within 150 ms intervals, voltages above 85 µV and below −85 µV, and a maximum voltage difference of less than 0.50 µV within 100 ms intervals. These intervals were rejected from individual channels in each trial in order to maximize data retention. The EMG signal was filtered offline using a 28–500 Hz IIR bandpass and 60 Hz notch filter, then rectified using a moving average procedure with a time constant of 20 ms (Cacioppo, Tassinary & Berntson, 2007). EMG artifact rejection consisted of voltages above 100 µV and below—100 µV.

Using previous protocol for analyzing the ERN (e.g., Saunders, Rodrigo & Inzlicht, 2016), the continuous EEG was segmented into epochs that commenced 200 ms before the response and lasted for 800 ms post-response. The ERPs (both the correct-related negativity; CRN, and ERN) were averaged separately for each experimental condition (ritual versus control). The CRN and ERN were operationalized as the mean amplitude between 0 and 80 ms at electrode FCz (e.g., Wiswede, Münte & Rüsseler, 2009). This time window was selected using the collapsed localizers approach suggested by Luck & Gaspelin (in press) to deal with the problem of multiple implicit comparisons inherent in ERP research. Though we used 0–80 ms based on this approach, the results were no different when alternative time windows were used, including 0–100 ms. ERN calculations were based on no fewer than five artifact-free error trials (Olvet & Hajcak, 2009a; Olvet & Hajcak, 2009b). EEG data were baseline corrected between −150 and −50 ms pre-response.

Statistical analysis

All the analyses were conducted using multilevel modeling (MLM) with the MIXED function in SPSS (v. 22.0). A 2-level multilevel model was used to account for within-subjects performance and ERN nested within individual participants while estimating a random intercept for each participant. We included all possible main effects and interactions in each model tested. We used an unstructured covariance matrix and the between-within method for estimating degrees of freedom. Effect sizes were estimated with semi-partial R2, (Rβ; Edwards et al., 2008).

For the behavioral data (mean RTs and choice error rates), the effect of condition (1 = ritual, −1 = control) was modeled as a function of round (1 = pre, −1 = post) and conflict level (1 = high conflict, −1 = low conflict). For the ERP and EMG analyses, the initial model was identical to the behavioral data, except conflict level was replaced by trial type. Here, trial type accounted for the correct-related negativity (CRN) and error-related negativity (ERN) ERPs (1 = CRN, −1 = ERN).

Results

Behavioral performance: choice error rates and mean RTs

All data are open and available on the Open Science preregistration site (https://osf.io/h64cz). We report all measures, manipulations, and exclusions. Confirming the task’s conflict manipulation, the model indicated a large main effect of conflict, with participants generating higher error rates on high-conflict trials (M = 20.16%, SE = 1.13) than low-conflict trials (M = 0.72%, SE = 1.13), b =  − 17.28, t(138) =  − 7.04, SE = 2.46, p < .0001, 95% CIs [−22.14–12.43], Rβ = .66. There was also a small, albeit non-significant effect of round, b = 4.13, t(138) = 1.68, p = .095, SE = 2.46, 95% CIs [−0.73–8.98], Rβ = .01, such that participants tended to have lower error rates in the pre rounds (M = 9.56%, SE = 1.13) than the post rounds (M = 11.29%, SE = 1.13). These analyses confirm the high versus low conflict manipulation. There was no main effect of ritual, b = 0.30, t(138) = 0.11, p = .92, 95% CIs [−5.25–5.85], Rβ2=0.001, nor was there a significant two-way interaction between ritual condition and round, b =  − 0.69, t(138) =  − 0.21, p = .84, 95% CIs [−7.28–5.91], Rβ = 0.001. Together, these results suggest that the ritual manipulation had no effect on error-rates.

Next, the same model was run with mean RTs. Again, the model indicated a large main effect of conflict, with participants responding more slowly during high-conflict trials (M = 414.44 ms, SE = 7.42) than during low-conflict trials (M = 346.09 ms, SE = 7.42), b =  − 65.48, t(138) =  − 9.33, SE = 7.02, p < .0001, 95% CIs [−79.36–51.60], Rβ = .75. There was no effect of ritual condition, b = 1.33, t(138) = 0.09, p = .93, 95% CIs [−29.81–32.47], Rβ = 0.001, nor of round, b =  − 1.86, t(138) =  − 0.27, p = .79, 95% CIs [−15.75–12.02], Rβ = 0.00, and no two-way interaction between ritual condition and round, b =  − 5.06, t(138) =  − 0.53, p = .60, 95% CIs [−23.92–13.80], Rβ = 0.001.

We note that we did not collect time data on individual participants’ self-paced rest breaks in between blocks. However, referring back to our observation and notes of the experiment, the average break between blocks was roughly between 5 s and 15 s. No participants took longer than 45 s–60 s. Future research should take this into consideration as it is important to assess the role of fatigue effects related to task performance across multiple trials.

Table 2 contains all the descriptive data for behavioral, self-report, and EEG measures. Theses analyses show that all participants were faster and more accurate in the low-conflict trials. However, there were no differences in performance between ritual and control conditions, suggesting that the ritual had no impact on behavioral performance.

Table 2 Means (SD) for self-reported subjective experience, performance on the inhibitory control task, and electroencephalography (EEG) measures.

Means across rows (within each condition) with different subscripts differ significantly at p < .05 (two tailed).

Dependent variable	Control condition	Ritual condition	
	Pre-rounds	Post-rounds	Pre-rounds	Post-rounds	
Positive emotion	4.76a (1.34)	4.76a (1.45)	4.86a (1.24)	4.73a (1.48)	
Efficacy	4.27a (1.30)	4.47a (1.42)	4.65a (1.27)	4.68a (1.02)	
Motivated preparation	4.90a (0.84)	4.70a (0.97)	5.08a (1.01)	5.13a (1.03)	
Positive ritual appraisal	–	5.33 (0.99)	
Low-conflict error rate (%)	1.0a (0.94)	0.58b (0.65)	0.84a (1.12)	0.46b (0.78)	
High-conflict error rate (%)	18.42a (12.17)	21.86b (14.58)	18.12a (12.83)	22.25a (15.06)	
Low-conflict reaction time	352.96a (52.22)	343.54b(55.60)	350.50a (37.81)	337.35b (32.91)	
High-conflict reaction time	417.31a (58.69)	410.38a (66.67)	415.98a (54.46)	414.11a (59.52)	
ERN (µV)	−2.26a (6.07)	−2.23a (5.97)	−4.87a (7.97)	−1.22b (6.09)	
CRN (µV)	4.27a (5.61)	3.73a (5.46)	5.21a (5.58)	3.81b (5.44)	
ΔERN (ERN-CRN)	−6.54a (7.55)	−6.05a (5.09)	−10.08a (7.55)	−5.07b (5.20)	

ERN data

We next tested our main hypothesis to examine whether arbitrary ritual impacts the ERN either by increasing performance-monitoring or decreasing performance-monitoring. To do this, the effect of condition on the error-related ERPs (ERN and CRN) was examined. The ERP data (i.e., the ERN and CRN while including the factor of trial type in the model; see above) was considered. The model output revealed evidence of a reliable ERN as indicated by a more negative ERP amplitude in response to erroneous (M =  − 2.65μV , SE = 0.86) than correct (M = 4.26μV , SE = 0.78) responses, b = 10.08, t(46) = 6.27, SE = 1.61, p < .0001, 95% CIs [6.84–13.31], Rβ = .63. This was qualified by a significant two-way interaction between round and trial type, b =  − 5.05, t(46) =  − 3.27, SE = 1.55, p = .002, 95% CIs [−8.16–1.94], Rβ = .14, indicating that the difference between ERN and CRN amplitudes was lessened in the post rounds (MERN =  − 1.73μV , MCRN = 3.77μV ) compared to the prerounds (MERN =  − 3.57μV , MCRN = 4.74μV ), an effect consistent with past work on the neural correlates of fatigue (Boksem, Meijman & Lorist, 2006; Inzlicht & Gutsell, 2007).

Critically, this two-way interaction was subsumed under a three-way interaction, where the interaction between round and trial type was moderated by our manipulation of ritual condition, b = 4.48, t(46) = 2.14, SE = 2.09, p = .037, 95% CIs [0.27–8.69], Rβ = .09. Figure 2 displays the ERP and topographical graphs.2 These show a relatively large reduction in the post rounds compared to the pre rounds for participants in the ritual condition. It appears, therefore, that a dampened ERN in the ritual condition offers support for the hypothesis that ritual decreases performance-monitoring, but without hindering overall performance.

Figure 2 Error-related ERPs as a function of condition assignment (A, control condition; B, ritual condition), trial type (correct, dotted lines; errors, solid lines) and manipulation time-point (pre-rounds, black lines; post-round, colored lines).

Spline head maps: scalp distributions of the ΔERN (difference wave = ERN-CRN) for mean activity (0–80 ms) as a function of condition assignment and time-point.

These analyses were followed up with a series of simple effects tests. First looking at participants in the ritual condition, the model revealed that during error trials, the ERN amplitude was significantly reduced (i.e., less negative) in the post rounds (M =  − 1.22μV , SE = 1.30) compared to pre rounds (M =  − 4.87μV , SE = 1.70), t(46.83) = 2.24, p = .03, d = 0.51). Moreover, opposite to the this pattern, it was found that during correct trials, the CRN amplitude was significantly less positive in the post rounds (M = 3.81μV , SE = 1.16) compared to the pre rounds (M = 5.21μV , SE = 1.19), t(45.30) = 2.95, p = .005, d = 0.25. The same comparisons done with participants in the control condition showed that the ERN amplitudes were no different in the pre rounds (M =  − 2.26, SE = 1.56) compared to the post rounds (M =  − 2.23, SE = 1.17), t(45.79) = 0.02), p = .98, d = 0.004. The CRN amplitudes were also no different during pre (M = 4.28, SE = 1.10) and post rounds (M = 3.73, SE = 1.07), t(45.05) = 1.26, p = .21,  d = 0.10. The simple effects comparing ERNs/CRNs within pre and post rounds revealed no differences between ritual and control conditions. CRNs for both conditions were no different for both pre rounds (t(46) = 0.58, p = .57, d = 0.17) and post rounds (t(46) = 0.04, p = .97, d = 0.01). There was also no differences in ERNs for both conditions in pre rounds (t(46) = 1.13, p = .26, d = 0.33) and post rounds (t(45.86) = 0.58, p = .57, d = 0.17).

Research typically indicates that a lower ERN relates to poor performance (e.g., Hobson et al., 2014), but this was not the case here. Looking at the ritual conditions separately, a dampened ERN was generally associated with worse performance, as revealed in higher error-rates, but only in the control condition (pre round: r(26) = .34, p = .09; post round: r(26) = .47, p = .015). This link did not hold for participants in the ritual condition (pre round: r(22) = .15, p = .51; post round: r(21) = .24, p = .29). These results suggest that the effect of a muted ERN after completing the ritual was somewhat independent of task performance. Though this is not directly tested here, it is possible that the ritual in this case facilitated performance by impacting the emotional component of the action-monitoring system (dulling the neuroaffective response to errors) without sacrificing features of conflict-monitoring and cognitive performance. However, more work is needed to directly test this hypothesis.

Altogether, the evidence from the above analyses offers modest support for the hypothesis that arbitrary, ad-hoc ritual (despite not being called ritual) reduces performance-monitoring, muting the neuroaffective response to distressing performance failures during task performance.

Spline head maps: scalp distributions of the ΔERN (difference wave = ERN-CRN) for mean activity (0–80 ms) as a function of condition assignment and time-point.

Self-report data

Self-reported emotion, efficacy, and motivation

Three separate models were run on the self-report data. The trial-type factor (error versus correct response) was removed from the model as we had no self-report data at this level of analysis. First, for positive emotion ratings, there was no main effect of condition, b =  − 0.08, t(45) =  − 0.21, SE = 0.40, p = .84, 95% CIs [−0.89–0.72], no main effect of round, b = 0.003, t(45) = 0.01, SE = 0.21, p = .99, 95% CIs [−0.43–0.43], and no interaction, b = 0.10, t(45) = 0.33, SE = 0.32, p = .75, 95% CIs [−0.53–0.74]. After completing the pre-rounds and post-rounds, there was no difference in positive emotions felt in both the ritual condition (Mpre-rounds = 4.79; Mpost-rounds = 4.68) and control condition (Mpre-rounds = 4.77; Mpost-rounds = 4.77). Similar patterns were found with efficacy ratings, with no main effect of condition (b = 0.29, t(45) = 0.76, SE = 0.37, p = .76, 95% CIs [−0.46, 1.03], no main effect of round, b =  − 0.18, t(45) =  − 0.94, SE = 0.19, p = .35), 95% CIs [−0.56–0.20], and no interaction, b = 0.09, t(45) = 0.33, SE = 0.28, p = .74, 95% CIs [−0.47–0.66]. Participants reported feeling no difference in efficacy in both the ritual condition (Mpre-rounds = 4.65; Mpost-rounds = 4.73) and control condition (Mpre-rounds = 4.27; Mpost-rounds = 4.45). Finally, the same pattern was found for ratings of motivated preparation, with no main effect of condition b = 0.42, t(45) = 1.52, SE = 0.28, p = .13, 95% CIs [−0.13–0.98], no main effect of round, b = 0.20, t(45) = 1.07, SE = 0.19, p = .29, 95% CIs [−0.18–0.58], and no interaction, b =  − 0.25, t(45) =  − 0.89, SE = 0.28, p = .38, 95% CIs [−0.81–0.31]. Again, participants’ motivated preparation was no different in both the ritual condition (Mpre-rounds = 5.08; Mpost-rounds = 5.13) and control condition (Mpre-rounds = 4.90; Mpost-rounds = 4.70). These analyses indicate that participants from both conditions self-reported no changes in affect/motivation going from time 1 to time 2.

Exploratory analyses: self-report emotion and appraisals, ERN, and performance

We next conducted a set of exploratory analyses that were planned after the initial pre-registration, namely exploring the relationship between our different observed variables: ERN, performance, and self-reported emotion/motivation. We also examined participants’ general positive appraisal of the ritual (it is important to note that since the questions were only given to participants in the ritual condition, these findings apply only to this group). Table 3 shows the correlations between all variables. For ease of interpretability, all variables (with exception to the positive ritual appraisal) were coded as a difference-score to represent the change in value from pre- to post-rounds (e.g., Efficacy = efficacypost–efficacypre). In this case, a more positive value would indicate an increase in efficacy form the pre- to the post-rounds. The correlations indicate that the ERN/ΔERN were unrelated to the self-report variables.

Table 3 Correlations between subjective experience of emotion/motivation, performance error-rates/reaction times, error-related neural monitoring, and positive ritual appraisals.

All variables represent the difference scores accounting for changes from pre-rounds to post-rounds.

Measure	1	2	3	4	5	6	7	8	
1. Motivated preparation	–	.25***	.02	.00	.22	−.21	−.20	.38***	
2. Positive affect		–	.51**	−.27***	.28***	−.13	−.11	.43*	
3. Efficacy			–	−.46**	.34*	−.05	−.03	.41***	
4. Error rate				–	−.73**	.06	.02	−.53*	
5. Reaction-time					–	−.22	−.08	.68**	
6. ERN						–	.96**	−.05	
7. ΔERN							–	−.06	
8. Ritual appraisal								–	
Notes.

* p < .05.

** p < .01

*** p < .10.

Looking within the ritual condition alone, we found that positive ritual appraisals were associated with both self-report and performance (but not ERN). Specifically, participants who evaluated the ritual experience as positive (i.e., meaningful, special, easy, etc.) were more likely: to feel better prepared heading into the post rounds; to commit fewer errors and respond more cautiously with longer RTs in the post rounds; and to report feeling more positive affect and greater efficacy after completing the post-rounds. Thus, it appears that the people who benefited from improved performance and heightened positive emotion/efficacy, were the ones who believed the actions actually had some sort of a positive impact on their performance.

These findings offer some clues to how rituals operate in the real-world and explain why these puzzling behaviors persist. Often the compulsion to do the ritual (and to do it properly) is because the person believes, for whatever reason, the ritual works to help achieve a desirable outcome. As this placebo effect builds across time, ordinary gestural actions start to develop personal meaning, transforming mere actions into a symbolic ritual practice.

Exploratory analyses: open-ended responses

Finally, we recorded these same participants’ open-ended responses to their impressions of the action sequences and their experience with them over the course of the week. Again, these responses were only collected for participants in the ritual condition. To investigate the content of the responses, we used the Linguistic Inquiry and Word Count 2007 (LIWC) software. LIWC is a text analysis program that counts words in a string of text and places them into preset psychological categories (Pennebaker, Booth & Francis, 2007). We looked at the four relevant LIWC categories, which included affective language (positive and negative valenced words) and self- versus other-directed language (self-referential and social words). One participant was not included in the analyses because they did not write anything in the response. Analyses revealed that participants used more self-referenced words (M = 10.98%, SD = 14.35) than other-referenced words (M = 2.03%, SD = 11.74), t(20) = 8.00, p < .0001, suggesting the ritual was a personal experience rather than a social one. Also, aligning with the above self-report findings, participants reported more positive emotion words (M = 3.84%, SD = 11.07) than negative emotion words (M = 0.44%, SD = 0.73), t(20) = 4.39, p < .0001.

Many of the responses illustrate the perceived benefits of the ritual and how it helped make participants feel a greater sense of calm during their performance. Interestingly, the feature of sequenced action repetition was frequently mentioned. For instance, one participant wrote: “I was able to observe that the repetition of activities somehow improved the completion of the tasks. I think that maybe completing the set of actions helped me feel a little more focused and calmed”. Another participant wrote: “Over the week, and in the study, completing the actions before beginning the task helped calm myself and make me feel in control for some reason”. The ritual appeared to be helpful even beyond the scope of the study and its tasks: “I found that by doing the action sequences at home, it made me more focused on my studies for the upcoming midterms”.

Tying this back into ritual’s effects on reduced performance-monitoring, a dampened ERN signal could be related to participants’ belief or perception that the ritual mitigated anxiety, and in turn, benefitted their overall performance. Taken together, across all the findings we find support that even arbitrary action sequences resembling ritual minimize performance-monitoring by dulling the neurophysiological reactivity to distressing self-generated failures, while still maintaining adequate performance.

Discussion

With a multiday experimental design using arbitrary and novel ritual-like actions, we offer causal evidence that performance rituals impact the neural regulatory system. Specifically, in a pre-registered experimental design, we put forward the question: Is ritual’s regulatory function driven by increases in performance-monitoring and motivated control (i.e., amplified ERN), or by decreases in performance-monitoring, which might relate to error-related distress (i.e., muted ERN)? The current findings offer more support for the latter: arbitrary ritual mutes the brain-based system attuned to the neuroaffective experience of regulatory failure. Despite decreases in performance-monitoring, however, the reduced ERN did not predict worse performance as it did in the control condition, a dissociation suggesting that ritual may be capable of regulating performance distress without sacrificing controlled behavior.

Indeed, recent research shows that a decrease in performance-monitoring is consistent with the experience of a person being less affected by self-generated errors and other sources of uncertainty/conflict. In the performance-monitoring domain, this pattern of a reduced ERN in response to performance failure is also consistent with other research highlighting the neurophysiological regulation of distress, anxiety, and uncertainty (e.g., Hajcak, McDonald & Simons, 2004; Jackson, Nelson & Hajcak, 2016; Tullett, Kay & Inzlicht, 2015). Further, the notion that rituals act as a palliative (consistent with a dampened ERN) aligns with similar research showing that religious beliefs act as a buffer against the pang of anxiety felt from self-generated failure (e.g., Good, Inzlicht & Larson, 2015; Inzlicht et al., 2009; Inzlicht & Tullett, 2010). Given the degree of overlap between ritual and religion—i.e., religion believed to be born out of the behaviors of early human ritual—it is reasonable to expect they share a common neural basis.

One implication of a general decrease in performance-monitoring is that since the detection of errors signals the need to initiate remedial action for better future performance, being less perturbed by failure (either through religious belief or ritual practice) could be considered maladaptive in certain contexts (e.g., Inzlicht, Bartholow & Hirsh, 2015; Inzlicht & Legault, 2014). As noted, however, this is not the case here. The fact that we do not see performance deficits linked to a reduced ERN might mean that ritual is capable of modulating performance-monitoring to ensure that an appropriate amount of error-related distress is still felt (but not too much). In other words, a ritual could help a person during performance find the anxiety sweet-spot that appears ideal for generating effective control (Yerkes & Dodson, 1908).

These findings complement the recent set of studies by Brooks et al. (2016) who found that arbitrary one-off rituals improve performance by reducing anxiety. The current findings build off this work in two main ways: first, we investigated rapid, online brain activity of the performance-monitoring system in response to the gold-standard of cognitive performance, a classic executive function task (e.g., Baddeley, 1996; Diamond, 2013). Second, we operationalized ritual as a week-long practice (versus done once in the lab) involving regular repetition, an important feature of real-life ritual (Legare & Souza, 2012). Indeed, our related work has shown that repetition is necessary in order for ritual to exert an influence on behavior (Hobson et al., in press).

More broadly, the growing interest in the experimental study of ritual (e.g., Lang et al., 2015; Schjoedt et al., 2013; Vohs et al., 2013; Wen, Herrmann & Legare, 2016) offers converging evidence that the even the most minimal and basic ritual actions (behaviors stripped of all personal and cultural meaning) can have predictive effects on brain and behavior. The value of such an empirical approach is that it can help both researchers and applied practitioners understand how performance rituals are created and for which purpose; that is, to understand the psychological processing that is involved as a ritual develops from being mere stereotyped actions to an important practice apart of one’s identity.

The current findings hint at a number of possible explanations. A person may ritualize an experience, for instance, by ascribing meaning to meaningless actions (Kapitány & Nielsen, 2015) and by automating predictable sequences over time. Though the ritual actions were initially framed as arbitrary, they likely developed a sense of purpose over the duration of the week. As hinted at in the self-report data, perhaps participants began to attribute their own personal meaning to the actions, making the connection between the completion of the ritual and achievement on the at-home task. The frequent practice of the actions would have also automated the motor sequence, which gets back at the idea of reinvestment and the extent to which conscious processing during a task disrupts motor performance (e.g., Masters, 1992). Rituals are comprised of overlearned, predictable action sequences that are easily automated with frequent practice, and as a result, may be useful in terms of preventing unwarranted conscious motor control—“reinvestment”—before and during performance (for a similar argument see Jackson & Masters, 2006). Thus, to the extent that seemingly arbitrary movements facilitate effective performance, certain rituals could over time become a functional tactic utilized by a performer in contexts where reinvestment could occur.

Limitations and future directions

First, the evidence in support of ritual’s palliative function comes mostly from ERN differences. Although we found arbitrary ritual dulled the neuroaffective system in response to error, there were no changes in self-reported affect to indicate that ritual made people consciously feel less anxious. Thus, it remains an open question whether ritual’s effects on a reduced ERN is related to anxiety regulation per se. It is possible that the reduced ERN corresponded to a psychological change independent of negative affect; this explanation should be weighed equally seeing that the affect-ERN link is not necessarily a universally accepted idea (e.g., Moser et al., 2013). Similarly, although participants’ performance on the inhibitory control task tended to become worse from pre to post, it did so for both groups. It appears then that ritual impacts the neural regulatory system, primarily the rapid emotion-based processing that occurs prior to conscious error awareness. Despite this, we find no effect of ritual on actual performance, which lies in contrast to the notion that one of the functions of this style of preparatory ritual is to facilitate effective task performance. Together it suggests that the current findings should be interpreted with some caution. To more fully understand ritual’s regulatory basis, more work is needed (survey, behavioral, and neuro) to determine when ritual impacts behavior and how this relates not only to underlying neurophysiology, but to the subjective experience of anxiety and motivated performance on a task. For instance, are there certain rituals in real life that are more likely to impact performance, and furthermore, are the rituals that fail to have an effect still carried out by the practitioner? Why?

Second, the altered performance-monitoring in post-rounds for the ritual condition occurred by a combination of a reduced (i.e., less negative) ERN and enhanced (i.e., more negative) CRN. Thus, ritual’s observed effects may not be exclusively a result of a dulled response to error (ERN), but a combination of an altered response to both errors and correct responses (both ERN and CRN). An enhanced CRN can occur when there is greater amount of conflict in the stimulus—response association, or when uncertainty in performance is high (Pailing & Segalowitz, 2004; Scheffers & Coles, 2000). In clinical and health research, this type of reversal pattern in ERN/CRN is thought to be an indication of a global disruption in performance-monitoring (e.g., Pietschmann, Simon & Endrass, 2008) and has been found particularly in cases of schizophrenia (Mathalon et al., 2002) and in aging adults (Endrass, Schreiber & Kathmann, 2012). Of course, it is unclear why we found this type of reduced dissociation patterning after the enactment of ritual. Whether ritual might actually hinder overall performance-monitoring is a question that cannot be addressed with the present data but should be investigated more thoroughly in future replications.

Third, returning to the pre-manipulation differences in ERN amplitude, we can see in Fig. 2 that the ritual group’s pre-round ERN appears much larger than that of the control group, which could account for the muted ERN effect in the ritual condition. However, as indicated in the analyses, the pre-manipulation ERN differences in the ritual condition did not differ statistically, making it difficult to disambiguate the driving factors leading to the observed changes. Given the multiday experience of the ritual manipulation at home, it is possible that participants experienced a boost in performance-monitoring at the start of the lab task.

Finally we note how the physical features of the arbitrary ritual could have led to the observed effects. Different rituals are made up of different physical movements. For instance, think about how a wedding ritual looks physically different than a funeral ritual. The physical features that we used to create the arbitrary ritual (i.e., slow, deliberate movements) had a meditative and calming feel to them, which involved frequent interval breathing. The muted ERN could be attributed to a physical state of relaxation as opposed to the ritual movements themselves. If the actions were faster and more action-oriented, like the pre-performance ritual of a pumped-up athlete or musician, then it is possible that we would have seen the opposite pattern of effects and an increase in performance-monitoring. We should therefore be cautious in generalizing the current findings to fit all types of rituals.

Conclusion

Taken together, the current experiment offers causal evidence of the psychological and neural basis of arbitrary ritual actions and their associated regulatory function. Though rituals might appear on the surface to be wasteful for expended energy and time, the presence of rituals in different performance contexts suggests they are critical to self-regulation and goal attainment. Here we offer preliminary support for the hypothesis that even an arbitrary ritual acts as a palliative by dulling the neural response to performance failure.

We thank B Saunders, H Lin, N Brown, Z Francis, A Ferguson, & L Zuber for helpful comments.

Additional Information and Declarations

Competing Interests

Author Contributions

Human Ethics

Data Availability

1 Superstitions are a style of thinking that have been linked to rituals (see Risen, 2016; Zhang, Risen & Hosey, 2014). A ritual may involve an element of superstitious thought like when the behavior of knocking on wood (ritual) is coupled with the belief that doing so pushes away bad luck (superstition). Of course, since superstitions are thought based, a person may have a superstition but without the overt ritual behavior or action. The key piece being that ritual is considered the coming together of symbolic thought with outward directed action (Geertz, 1973).

2 Upon visual inspection of the ERP graphs, it appears there might be an effect of the Pe–a later, slower moving error-related waveform that is thought to be tied to the conscious awareness of error commission. To explore this, we conducted similar analyses looking at the Pe as defined as the average mean amplitude between 200 and 400 ms following responses on error trials at the posterior electrode site, Pz. The model revealed a significant main effect of trial type, b = 9.50, t(46) = 4.89, SE = 1.94, p < .00001, 95% CIs [5.59–13.40], indicating that the Pe was larger for error responses (M = 8.29μV , SE = 1.35) compared to correct responses (M =  − 2.68μV , SE = 1.02). There were no other significant main effects or interactions (all ps > .15).

The authors declare there are no competing interests.

Nicholas M. Hobson conceived and designed the experiments, performed the experiments, analyzed the data, contributed reagents/materials/analysis tools, wrote the paper, prepared figures and/or tables, reviewed drafts of the paper.

Devin Bonk performed the experiments, analyzed the data, contributed reagents/materials/analysis tools, wrote the paper, prepared figures and/or tables, reviewed drafts of the paper.

Michael Inzlicht conceived and designed the experiments, contributed reagents/materials/analysis tools, wrote the paper, reviewed drafts of the paper.

The following information was supplied relating to ethical approvals (i.e., approving body and any reference numbers):

The University of Toronto granted ethical approval to carry out the study within its facilities (protocol reference #28411).

The following information was supplied regarding data availability:

Hobson NM (2017). Data. Available at https://osf.io/h64cz.

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
