# Peer review of "Rituals decrease the neural response to performance failure"

_PeerJ, doi:10.7717/peerj.3363_

## Round 0.1 · original submission · Major Revisions

I now have received three reviewers' comments. Although all reviewers have expressed their interest in your study, several aspects of this manuscript should be revised to improve its clarity. Their observations are presented with clarity so I'll not risk confusing matters by belaboring or reiterating their comments. While I might quibble with the occasional point, I note that I regard the reviewers' opinions as substantive and well-informed. I believe that all of the highlighted reservations require contemplation and appropriate attention in revising the document if it is to contribute appropriately to Peer J and the extant literature. Please revise or refute according to the two reviewers' comments and provide a point by point reply in addition to the revised manuscript.

Tsung-Min Hung, Ph.D.
PeerJ editor
Distinguished professor
Department of Physical Education
National Taiwan Normal University

Reviewer 1 ·

Basic reporting

Well-written paper with clear hypotheses and appropriate reference to background literature.

Experimental design

Generally clear, and good work in pre-registering the experiment. Information on the duration of the self-paced rest for both groups should be included to allow one to determine the role of fatigue effects in future studies. Most importantly, it is essential to clearly discuss the manipulation check for the study. It is a very complex design and it is unclear whether the participants adhere to the experimental protocol while at home.

Validity of the findings

Novel study with robust descriptive and inferential analysis.

Excessive speculation in the Discussion. Greater dialogue with the previous empirical studies is needed in a revised version of the manuscript.

Additional comments

There is much to like about this paper but there is also much room for improvement. Below, I have provided constructive feedback to the authors.

1 - In the Introduction, the authors need to differentiate rituals from pre-performance routines. There is extensive research on pre-performance routines in the performance and sport psychology literature.

2 - The authors wrote: "payed attention to the chunked a predictable action sequences of a personal ritual can act a compensatory strategy" that aids performance. However, there is extensive research claiming exactly the opposite (see theory of reinvestment).

3 - There are many technical terms used loosely in the paper ("motivational processes"; "personal mastery"). This needs to be reconciled. If a psychological term is referred to, then a clear explanation of how this matters to this study (how was it measured?) and the theoretical framework behind it (e.g, is personal mastery related to self-efficacy or motivation climate?!) should be provided.

4 - It is important to include effect size estimates for all comparisons. Table 2 could be turned into a Figure instead.

5 - Provide the reader with a clear section or sub-heading for the "manipulation checks" used in this study. The experimental protocol is complex and it is unclear whether the home part of the experiment was properly completed by the participants .

6 -Information on the duration of the self-paced rest for both groups should be included to allow one to determine the role of fatigue effects in future studies.

7 - The Discussion section is way too speculative. The authors themselves use the word speculation and its variation many times. Greater connection to previous theory and empirical studies is needed.

Reviewer 2 ·

Basic reporting

Minor point: Line 126: ‘Avery’ should be corrected to ‘A very’

While the manuscript is well written and methods described in great detail, I feel it is extremely long for a single figure paper. The authors would be able to better highlight their main finding to the readers if the manuscript were to be condensed.
The experiment has been well designed and the analysis is extremely thorough.

Experimental design

The experiment has been well designed and the analysis is extremely thorough.

Validity of the findings

The study presents an interesting finding, suggesting that rituals tend to reduce performance monitoring without affecting the actual performance. Overall the study is well performed and analyzed. The authors provide sound interpretation backed by their data and by explicitly suggesting limitations of their study, the authors help readers (including me) better evaluate their work.
The statistical analysis is rigorous and sound.

·

Basic reporting

The manuscript is written well, flows and clear. All sections are thorough, detailed and informative. References are up-to-date and extensive. The analyses conducted is related to the hypotheses. Only a few minor general comments
1) The distinction among rituals, routines, habits and even superstitions needs to be expanded in the literature review section.
2) CRN is not discussed in the literature review, even though it is measure and is included in the results and discussion section. I think it should first be introduced in the literature review section.
In addition, some specific comments:
1)Line 199: have double "((" delete one.
2) Line 214: Spell out ERP in first instance
3) Line 233: add "of" so that it reads "...a final sample of..."

Experimental design

The study is original and innovative in that it examines the underlying mechanisms of rituals from a neuropsychological perspective. Rituals are performed by most people across a variety of tasks (everyday and domain specific). The research questions are clear regarding the effects of ritual behaviours on ERN (increase or decrease). The design is experimental, with a figure to help explain the procedure of the testing in the lab.

Validity of the findings

Findings are valid and the results and discussion sections are linked. The results section is comprehensive and organised well addressing each hypothesis and some exploratory analyses as well. The discussion highlights the main findings and limitations are included and are honest and thorough. The only comment is that results indicated that performance was not linked with rituals, ERN, CRN in anyway. The assumption was that rituals "regulate goal-directed performance" but in the study this was not found (at least by a performance outcome measurement - RT and accuracy). This limitation although mentioned, should be emphasized.

Additional comments

Enjoyed reading the manuscript. Was easy to follow and understand. The study is novel and interesting.

---

## Round 0.2 · Major Revisions

I now have received both reviewer's comments on your revised manuscript. Generally the reviewers were positive about your revision. However, one of the reviewers has pointed out several aspects of your manuscript that require remaining revision. Please revise or refute according to the reviewer's comments and provide a point by point reply in addition to the revised manuscript.

Tsung-Min Hung, Ph.D.
PeerJ editor
Distinguished professor
Department of Physical Education
National Taiwan Normal University

Reviewer 1 ·

Basic reporting

Previously addressed.

Experimental design

Previously addressed.

Validity of the findings

Previously addressed.

Additional comments

Overall, I commend the authors for addressing most of the requested suggestions. There are still issues that need to be resolved before the paper reaches its full potential.

1. The tile is too bold given the authors advise to take their findings with caution (e.g., "Together it suggests that the suggests the current findings should be interpreted with some caution"). Also, as per the response letter, the authors have noted that the discussion is primarily focused on the limitations of the study. As such, the title should reflect a more conservative tone.

2. I maintain that the Table 2 is not very telling. A graph with error bars will be easier to grasp and the table can go into the supplementary materials to allow detail information for meta-analysists and scholars interested in replicating the findings.

3. Information about fatigue effects are not of marginal importance to the experimental protocol and are crucial to inform future research as well. As such, the role of fatigue effects should be discussed in the main body of the text rather than added as an ancillary "note" at the end of the manuscript.

4. The reference to Reinvestment Theory in the discussion is not well-developed. I suggest the addition of another sub-section in the Discussion, wherein the authors develop the theoretical and applied implications of their findings. The authors have now navigated many theoretical concepts (self-efficacy, reinvestment, rituals) but have focused their discussion in none. Theoretical integration and development is at the core of open science, and the notion of pre-registered experiments.

·

Basic reporting

The authors have addressed the general and specific comments in this section. Specifically, they added a not concerning superstitions and a more thorough discussion of the differences between rituals and routines in the introduction section.

Experimental design

No comments to address from the first review.

Validity of the findings

The authors have addressed the general comments in this section that the findings did not indicate significant RT and accuracy outcome changes. They added information in the discussion section to emphasize this point.

Additional comments

Well done addressing all the comments. The paper reads better and you have addressed all the comments.

---

## Round 0.3 · accepted · Accept

I have now received the first reviewer's comment with satisfactory feedback. You and your coauthors have my congratulations. Thank you for choosing PeerJ as a venue for publishing your research work and I look forward to receiving more of your work in the future.

Tsung-Min Hung, Ph.D.
PeerJ editor
Distinguished professor
Department of Physical Education
National Taiwan Normal University

Reviewer 1 ·

Basic reporting

Addressed.

Experimental design

Addressed.

Validity of the findings

Addressed.

Additional comments

I commend the authors for taking on board the constructive feedback and addressing in full all requested revisions.